# An Ultra-Stretchable Polyvinyl Alcohol Hydrogel Based on Tannic Acid Modified Aramid Nanofibers for Use as a Strain Sensor

**DOI:** 10.3390/polym14173532

**Published:** 2022-08-28

**Authors:** Lei Miao, Xiao Wang, Shi Li, Yuanyuan Tu, Jiwen Hu, Zhenzhu Huang, Shudong Lin, Xuefeng Gui

**Affiliations:** 1School of Materials Science and Hydrogen Energy, Foshan University, Foshan 528000, China; 2Guangdong Key Laboratory for Hydrogen Energy Technologies, Foshan 528000, China; 3Guangzhou Institute of Chemistry, Chinese Academy of Sciences, Guangzhou 510650, China; 4University of Chinese Academy of Sciences, Beijing 100049, China; 5Guangdong Provincial Key Laboratory of Organic Polymer Materials for Electronics, Guangzhou 510650, China; 6CAS Engineering Laboratory for Special Fine Chemicals, Guangzhou 510650, China; 7Incubator of Nanxiong CAS Co., Ltd., Nanxiong 512400, China

**Keywords:** aramid nanofiber, hydrogel, tea stain-inspired chemistry, mechanical property, strain sensor

## Abstract

The mechanical performance is critical for hydrogels that are used as strain sensors. *p*-Aramid nanofiber (ANF) is preferable as an additive to the reinforce the mechanical performance of a poly(vinyl alcohol) (PVA). However, due to the limited hydrogen bond sites, the preparation of ultra-stretchable, ANF-based hydrogel strain sensor is still a challenge. Herein, we reported an ultra-stretchable PVA hydrogel sensor based on tea stain-inspired ANFs. Due to the presence of numerous phenol groups in the tannic acid (TA) layer, the interaction between PVA and the ANFs was significantly enhanced even though the mass ratio of TA@ANF in the hydrogel was 2.8 wt‰. The tensile breaking modulus of the PVA/TA@ANF/Ag hydrogel sensor was increased from 86 kPa to 326 kPa, and the tensile breaking elongation was increased from 356% to 602%. Meanwhile, the hydrogel became much softer, and no obvious deterioration of the flexibility was observed after repeated use. Moreover, Ag NPs were formed in situ on the surfaces of the ANFs, which imparted the sensor with electrical conductivity. The hydrogel-based strain sensor could be used to detect the joint movements of a finger, an elbow, a wrist, and a knee, respectively. This ultra-stretchable hydrogel described herein was a promising candidate for detecting large-scale motions.

## 1. Introduction

Strain sensors are devices that can detect the deformations of a subject via signals (usually electric). A stretchable rather than a rigid material with the stretchability to exceed 300% is desirable for the preparation of strain sensors [1]. The ability to undergo high and reversible stretching deformation is one of the critical prerequisites for strain sensors that are used to monitor the large-range movements of human limbs [1,2,3,4]. In comparison with textiles or traditional elastomeric substrates, hydrogels are more promising candidates for the preparation of strain sensors because they can undergo a larger range of deformations and they are more similar to human skin due to their flexibility and biocompatibility [1,5].

Polyvinyl alcohol (PVA) is an inexpensive and water-soluble polymer with excellent biocompatibility [6] and has gained much attention as a precursor for the preparation of hydrogel-based strain sensors. However, it is widely known that PVA-based hydrogels are often too brittle for use in strain sensors [7]. To date, two methods have been reported that can be employed to improve the mechanical performance of a PVA-based hydrogel. For example, glutaraldehyde is used as a cross-linker to react with the hydroxyl groups (O-H) in PVA molecules [8,9]. However, a cross-linker would not only enhance the hydrogel’s tensile or compressive strength, it would also cause the hydrogel to become rigid. Unfortunately, a hydrogel with a rigid network would have a tensile modulus that does not match that of human skin, making it unsuitable for use as a strain sensor [2]. The other promising method is to add additives to enhance the physical interactions between PVA chains. The reported additives included other polymers or chemicals such as polyacrylic acid (PAA) [10], polyacrylamide (PAM) [11], borax [12], carrageenan [13] et al., and nanofillers such as cellulose nanofibrils [14], silk fibroin [15], or PVA nanoaggregates produced via “freezing-thawing” processes [16,17]. Based on this method, the physical interactions, such as the physical entanglement between polymer chains, hydrogen bonds, electrostatic attractions, and so forth, were strengthened, and the hydrogel’s stretchability was significantly improved. However, it is well known that a conductive network is necessary for a strain sensor [18], while the aforementioned additives or nanofillers cannot be directly used to fabricate PVA-based hydrogel sensors due to their electrical insulation. Electrically conductive additives, i.e., Ag nanowires [19], carbon nanofibers [20], etc., or carriers that can load conductive materials, i.e., Ag nanoparticles (Ag NPs) [21], polyaniline (PANi) NPs [22], etc., are required in a hydrogel matrix, which imparts the hydrogel with electrical conductivity.

Aramid nanofibers (ANFs), due to their outstanding physicochemical stability, have recently attracted much attention as building blocks of polymer composite materials with high mechanical performance [23], such as battery separators [24], filters [25,26], and biomimetic devices [27]. Due to hydrogen bonds between the amide groups (-NH-CO-) of the ANFs and the hydroxyl groups (-O-H) of PVA, ANFs could not only be used to provide mechanical reinforcements to PVA hydrogels [28], but can also serve as carriers to load conductive materials. Jia et al. recently reported a PVA/ANFs/PANi nanocomposite hydrogel sensor that was reinforced by ANFs [22]. They demonstrated that hydrogen bonds and the π–π stacking between aramid and PANi molecules enabled the deposition of PANi nanoparticles on the surface of ANFs. However, due to the limited hydrogen bond between ANFs and PVA, the tensile elongation of the PVA/ANFs/PANi hydrogel was only 140%.

To further improve the stretchability of the ANF-based PVA hydrogel strain sensor, we discerned that tannic acid (TA) was a very promising candidate due to the abundant -OH groups in a TA molecular structure. According to the report from Wang et al., PVA/TA composite hydrogels had high tensile elongations that exceeded 1100% [29]. In this work, we attempted to use tannic acid (TA) to modify ANFs via tea stain-inspired chemistry and subsequently use the modified ANFs as fillers to improve the mechanical properties of PVA hydrogels so that they could be used in strain sensors. TA, as the major component in tea-stains, has solid–liquid interfacial activity, and can create numerous active phenolic hydroxyl groups (Ph-OH) on the surfaces of various substrates [30,31]. We hypothesized that TA could be readily deposited onto the surfaces of ANFs and provide more hydroxyl groups (-O-H), so that the hydrogen bonds between ANFs and PVA could be remarkably strengthened. Moreover, Ag NPs could be formed in situ on the surface of the TA layer, and Ag NPs-loaded ANFs could provide a conductive network in the hydrogel matrix. With the use of these sensors, human movements could be monitored by detecting changes in the electrical resistance. This strategy might be useful for preparing PVA hydrogel-based strain sensors with high mechanical performance. The details and results in this work might provide a valuable insight enabling the preparation of novel ANF-containing composites with various applications.

## 2. Materials and Methods

### 2.1. Materials

Kevlar@49 threads were purchased from Dupont^TM^ (Wilmington, DE, USA). Prior to use, these threads were cut into pieces with a length of ~5 mm and washed, respectively, with acetone, ethanol, and distilled water in an ultrasound, before being subsequently dried at 80 °C for 24 h under vacuum. Polyvinyl alcohol (alcoholysis degree: 98–99%), tannic acid (TA), AgNO_3_, potassium tert-butoxide (KO*t*Bu), and other AR reagents were purchased from Aladdin Reagent (Shanghai, China) and used as they were received.

### 2.2. Preparation of PVA/TA@ANFs/Ag Hydrogel

**Aqueous ANF dispersion.** The neat ANF dispersion was prepared via the method described in our previous work [26]. In summary, to prepare an aqueous ANF’s dispersion, an ANF dispersion in DMSO with a concentration of 0.2 wt% was diluted to a final concentration of 0.1 wt% via the dropwise addition of an equal volume of DI water. Once the water was added, the dark red dispersion became cloudy and pale yellow. After the dilution, the resultant pale yellow dispersion was stirred for 2 h to form a stable ANF dispersion. Prior to further use, the obtained ANFs were thoroughly washed with water and separated via filtration to remove residual KO*t*Bu, methanol, and DMSO.

**TA@ANFs.** To prepare TA-modified ANFs (denoted as TA@ANFs), 1.0 M of *tris* buffer solution was added dropwise into 100 mL of an aqueous ANF dispersion with a concentration of 0.1 wt% until the pH was increased to ~8.5. Subsequently, 0.1 g of TA was added, and the dispersion was stirred for 12 h at room temperature. The TA@ANFs in the resultant yellow dispersion were gathered by centrifugation, and fresh DI water was added to the solid residue to obtain an aqueous TA@ANFs dispersion.

**PVA/TA@ANFs’ hydrogel.** PVA powder (8.82 g) was added into 50 mL of DI water or aqueous TA@ANFs dispersions with various ANF concentrations. The mixture was stirred for 5 h at 95 °C in order to ensure that the PVA had dissolved. Subsequently, the mixture was poured into a ribbon-shaped Teflon mold with dimensions of 115 mm × 40 mm × 12 mm and stored at −20 °C for 12 h. The frozen strip was then thawed and stored at room temperature for another 12 h. The PVA/ANFs hydrogel was obtained after three “freezing—thawing” cycles.

**PVA/TA@ANFs/Ag hydrogel.** Briefly, a 5.0 wt% aqueous NH_3_·H_2_O solution was added dropwise into a 0.5 wt% aqueous AgNO_3_ solution under stirring, and the stirring was continued until the solution became completely clear. Then, various PVA/ANFs hydrogels were immersed into the resultant aqueous Ag(NH_3_)^2+^ solution for 12 h to form Ag NPs in situ. After thorough washing with DI water, black PVA/TA@ANFs/Ag hydrogels that were denoted as PTAA-x were obtained, where x represented the mass of TA@ANFs in the hydrogel, which was 0, 5, 10, 15, 20, or 25 mg, respectively. The mass ratios of TA@ANFs in various PTAA hydrogels were 0.6‰, 1.1‰, 1.7‰, 2.3‰, and 2.8‰, respectively.

### 2.3. Structural Characterization

**Morphology observation.** The morphologies of the ANFs and TA@ANFs were observed with a JEM-100CX II transmission electron microscope (TEM, JEOL, Tokyo, Japan) with an acceleration voltage of 100 kV. Prior to observation, the aqueous ANFs’ or TA@ANFs’ dispersion was sprayed onto a carbon film-coated copper grid and dried under a flow of nitrogen. The morphologies of the hydrogels were observed via field emission scanning electron microscopy (FESEM) using an S-4800 (Hitachi, Tokyo, Japan) system with an accelerating voltage of 2.0 kV. All of the samples were freeze-dried and scattered with Pd for 30 s prior to observation.

**Chemical characterization.** The elemental compositions of the ANFs and TA@ANFs were analyzed via X-ray photoelectron spectroscopy (XPS, ESCALAB 250, Thermo Fisher Scientific, Waltham, MA, USA) at a base pressure of 10^−8^ mbar. The samples were irradiated with a monochromatic Al Kα X-ray source (1486.6 eV) at 100 W. A passing energy of 150 eV was used to record the survey spectra. An iS 50 system (Thermo Fisher) was used to characterize the chemical bonds in the ANF and TA@ANF samples via attenuated total reflectance Fourier-transform infrared spectroscopy (ATR-IR). The silver contents in the PVA/TA@ANFs/Ag hydrogels were analyzed via energy dispersive spectrometry (EDS) with the use of the FESEM system mentioned above.

### 2.4. Properties Measurements

**Mechanical properties.** The tensile strength was evaluated by using a CMT 7503 Universal Testing Machine (Shenzhen SANS, Shenzhen, China) with a tensile rate of 50 mm·min^−1^. Prior to the test, each sample was cut into ribbons with dimensions of 50 mm × 9 mm × 12 mm, and the distance between the two clamps of this device was fixed at 10 mm. The tensile stress (*σ*) was calculated via Equation (1):(1)$$\sigma =\frac{F}{S}$$
where *F* was the force recorded by the instrument and *S* was the cross-sectional area of the samples perpendicular to the stretch direction. The tensile strain (*ε*) was calculated via Equation (2):(2)$$\varepsilon =\frac{L}{L_{0}}\times 100\%$$
where *L* was the deformation length and *L*_0_ was the original length.

The compressive strength was evaluated by using the same instrument with a compression rate of 50 mm·min^−1^. The tests were stopped once the compression deformation reached 80%. Prior to the test, each sample was cut into a cylindrical shape with a diameter and a height of 25 and 30 mm, respectively. The compression stress (*δ*) was calculated via Equation (3):(3)$$\delta =\frac{F}{A}$$
where *F* is the force and *A* denotes the original cross-sectional area. The compression strain (*ε*′) was calculated via Equation (4):(4)$$\varepsilon^{\prime }=\frac{H}{H_{0}}\times 100\%$$
where *H* and *H*_0_ were the deformation height and the original height of samples, respectively. All of the measurements were performed five times, and the average value was recorded.

**Sensing measurements.** The ribbon-shaped hydrogel with dimensions of 10 mm × 25 mm × 1 mm was pasted onto the skin, covering fingers, elbows, wrists, and knees by adhesive tape. The electrodes of an inductance–capacitor–resistance (LCR) meter were respectively pasted at the contrary ends of the hydrogel along the deformation direction. The distance between these two electrodes was 18 mm. The relative resistance ratio (*R*_R_) was calculated via Equation (5):(5)$$R_{R}=\frac{R-R_{0}}{R_{0}}\times 100\%$$
where *R* was the resistance recorded by the LCR meter and *R*_0_ was the initial resistance of the sample without any deformation.

**Swelling kinetics.** Each hydrogel was immersed into DI water for 48 h and collected at each scheduled time point in sequence. The surface of the hydrogel was wiped with filter paper, and the mass of this hydrogel (*m*) was subsequently recorded. The swelling ratio (SR) was calculated via Equation (6):(6)$$\mathrm{SR}=\frac{m-m_{0}}{m_{0}}\times 100\%$$
where *m*_0_ was the initial mass of the dry hydrogel.

**Dewatering kinetics.** The hydrogel was immersed into DI water for enough time to absorb sufficient water so that its mass became constant. Subsequently, the swollen hydrogel was placed in a sealed dryer with a constant temperature of 35 °C and a relative humidity of 60% for 7 days. During this period, the mass (*m*) of the hydrogel was recorded at scheduled time points. The mass variation ratio (*m*_R_) was calculated via Equation (7):(7)$$m_{R}=\frac{m-m_{0}}{m_{E}-m_{0}}\times 100\%$$
where *m*_E_ was the mass of the hydrogel that had absorbed water and become swollen, while *m*_0_ was the mass of the dry hydrogel.

**Thermal curves.** The exothermic curves of the hydrogels were recorded via differential scanning calorimetry (DSC Q2000, TA instruments, New Castle, DE, USA). These measurements were recorded at a cooling rate of 5 °C/min over a temperature range from 40 to −80 °C under nitrogen protection.

## 3. Results

### 3.1. Tea Stain-Inspired, Chemistry-Based ANFs

The illustrations depicting the preparation of the tea stain-inspired, chemistry-based ANFs are shown in Figure 1. According to our previous work, DMSO/KO*t*Bu could capture the protons of amide groups and disrupt their hydrogen bonds. The resultant ANFs were stable in DMSO due to the solvation effect and the electrostatic repulsion between anionic amides (Figure 2A) [25]. To further modify the ANFs via tea stain-inspired chemistry, water, rather than a highly polar organic solvent, was required for the dispersion of ANFs because the auto-oxidized tannic acid layer was more stable in aqueous media than in acidic, alkaline, or organic solvents. Notably, the stability of this tannic acid layer was similar to that of polydopamine, another bio-inspired chemical reagent [32]. Three critical steps were used in this work to prepare water-stable ANFs. First, in contrast with our previous work, water, rather than hydrochloric acid, was used to return protons to amide groups. The protonation of anionic amides was delayed, which impeded the formation of hydrogen bonds between different ANFs. Second, water (which is a poor solvent for ANFs) had to be added dropwise to prevent the aggregation of ANFs. Third, a low concentration of ANFs (0.1 wt%) was employed to facilitate the dispersal of the ANFs.

As demonstrated by the photograph in Figure 2B, well-dispersed ANFs were obtained after they had been placed in water for 24 h. To further validate the dispersity of ANFs in aqueous media, we compared the diameters of ANFs that had been respectively dispersed in DMSO and water. As shown in Figure 3A,B, the diameter of the ANFs that had been dispersed in DMSO was 15 ± 9 nm, while that of the ANFs that had been dispersed in water was 21 ± 9 nm. Therefore, the diameter increased by a factor of less than 2.0 when DMSO was replaced by water, which indicated that there was no significant formation of aggregates in the dispersion. This result demonstrated that water-stable ANFs with good dispersity were obtained in this work.

Other researchers have proven that neat ANFs could form hydrogen bonds with the hydroxyl groups of PVA, thus helping to enhance the mechanical properties of PVA-based hydrogels [33]. However, aramids are poorly miscible with PVA because phase separation and self-agglomeration could be observed when the PVA matrix had a high aramid filler content [34]. To further promote the formation of hydrogen bonds between ANFs and PVA, and thus prepare PVA hydrogels with outstanding mechanical performance, we used tea stain-inspired chemistry to create more donor and acceptor sites for the formation of hydrogen bonds while also enhancing the hydrogen bond energy. TA can become deposited onto the surfaces of various polar or non-polar substrates via auto-oxidation [35].

Two findings demonstrated that TA molecules had indeed formed deposits on the surface of ANFs. First, as shown in the TEM images in Figure 3B, the diameter of neat ANFs in aqueous was 21 ± 9 nm, while the diameter of TA@ANFs was 25 ± 9 nm (Figure 3C), which revealed that the thickness of the TA layer was ~4 nm. Second, the elemental compositions of neat ANFs and TA@ANFs (with regard to C, N, and O) were respectively recorded by XPS (Table 1). Because the survey depth of XPS is usually less than 10 nm [36], it could be observed that the O content had slightly increased, and that the N content had significantly decreased after the deposition of TA, while the C content in TA@ANFs and neat TA did not change significantly. Thus, the surface of the ANFs could be considered to be adequately coated by the TA layer. Due to the excellent hydrophilicity of the TA layer, the aqueous TA@ANFs’ dispersion was very stable, and no aggregates were observed after 24 h of storage, as is demonstrated by the photograph in Figure 2C.

### 3.2. Preparation of the PVA/TA@ANFs/Ag Hydrogel

Crystalline and amorphous phases usually coexist in commercial PVA powder, and hydrogen bonds facilitate the formation of a crystalline phase in solid PVA [37]. It is widely known that hydrogen bonds will be weakened when the temperature is increased [38]. Thus, a temperature of 95 °C was used to disrupt the hydrogen bonds between PVA molecules and facilitate the preparation of the PVA solution in this work. Based on the following two considerations, a “freezing-thawing” strategy was used to prepare PVA/TA@ANFs’ hydrogels. First, a low temperature facilitated the formation of hydrogen bonds. We assumed that more hydrogen bonds could form to reinforce the interactions between PVA and TA@ANFs. Second, as has been reported by other researchers, a freezing–thawing cycling process promoted the formation of nano-crystalline PVA [39]. We assumed that the hydrogel’s network would be reinforced by the resultant nano-crystalline PVA. We observed that the PVA/TA@ANFs’ hydrogels that were subjected to three or more cycles of freezing–thawing had desirable mechanical properties, which may be attributed to the formation of more nano-crystalline. FTIR data provided confirmation of the formation of hydrogen bonds. As shown in Figure 4, strong absorption peaks at 2908 and 2937 cm^−1^ were ascribed to the stretching vibration of the methylene groups (-CH_2_-) in PVA. In addition, a stretching vibration peak corresponding to the hydroxyl groups (O-H) in the PVA’s crystalline regions was observed at 3250 cm^−1^, which shifted to 3284 cm^−1^ in the PVA/TA@ANFs’ hydrogel. The blue-shift of the O-H vibration indicated that the initial crystallization of PVA was disrupted and that new hydrogen bonds had formed between the PVA and TA@ANFs [40]. SEM observation is a facile method used to estimate the miscibility between PVA and TA@ANFs. As demonstrated by the SEM images of PVA/TA@ANFs’ hydrogels shown in Figure 5A–F, no individual ANF was observed after three “freeze-thaw” cycles, and the TA@ANFs were well dispersed on the PVA substrate, which indicated that there was excellent miscibility between the ANFs and PVA. However, the surfaces of the PVA/TA@ANFs’ hydrogels shown in Figure 5E,F were wrinkled rather and smooth, which might have been due to the differences in the diameters between the ANFs and the PVA polymer chains.

As common polyols, PVA and TA are often used as reducing agents to promote the conversion of Ag(I) to Ag(0) [41,42]. Ag NPs were not only formed on the PVA substrate, but also on the surfaces of TA@ANFs. However, due to the delocalization of π electrons in the neighboring hydroxyl group of polyphenol, the reducing activity of TA was stronger. TA@ANFs were an essential prerequisite for the construction of a conductive network of Ag NPs. A more continuous Ag NP-based conductive layer would be preferentially formed on the surfaces of the TA@ANFs rather than the formation of individual Ag NPs on the PVA substrate. The Ag content in our hydrogel was approximately 1.99 wt%, based on the EDS data. Moreover, due to its outstanding reducing activity, the TA layer can inhibit the oxidization of Ag NPs [43], which may enhance the durability of the PVA/TA@ANFs/Ag hydrogel.

### 3.3. Mechanical Properties of the PVA/TA@ANFs/Ag Hydrogels

Improving the stretchability of PVA hydrogels has the potential to significantly broaden their applications because some large-scale movements may damage the currently existing hydrogels and cause the formation of undesirable cracks [12,44]. The most effective strategy to enhance the stretchability is to strengthen the interactions between different polymer chains via various chemical bonds or physical effects. In this work, hydrogen bonds between TA and PVA were used to enhance the stretchability of PVA hydrogels. Based on the following two considerations, TA was selected in this study as a promising candidate with which to improve the mechanical properties of PVA hydrogels. First, each TA molecule includes 25 phenolic hydroxyl groups (Ph-O-H) that can act both as hydrogen bond donors and acceptors. Meanwhile, the 10 ester groups (-COO-), as well as the 1 ether (-O-) group in each TA molecule, can act as hydrogen bond acceptors. Each repeat unit in a *p*-aramid chain includes two carbonyl (C = O) groups that can act as hydrogen bond acceptors, as well as two amide groups that can act both as hydrogen bond donors and acceptors. We calculated the number of the sites in TA that can form hydrogen bonds, which was 2.12 per 100 g/mol of *M*_w_, while the corresponding value for *p*-aramid was 1.68 per 100 g/mol of *M*_w_. Second, the phenolic hydroxyl groups of TA could form more stable hydrogen bonds. Because length of the C-O bond in the phenolic hydroxyl groups was shortened due to the delocalization of π electrons, which shortened the distance between the oxygen atom in the phenolic hydroxyl group (Ph-O-H) and the hydrogen atom in other molecules, the hydrogen bond energy was thus enhanced [45]. On the other hand, the nano-crystalline PVA that was constructed by freezing–thawing could also promote the mechanical reinforcement, as has been demonstrated by other researchers [39,46]. Therefore, we assumed that the tensile strength of the PVA/TA@ANFs/Ag hydrogels would be synergistically enhanced by TA@ANFs and nano-crystalline PVA. When a tensile force was exerted on the hydrogels, they could share this tensile force to prevent the hydrogel from forming undesirable cracks.

Our hypothesis was confirmed based on the strain–stress curves of various PVA/TA@ANFs/Ag hydrogels. As shown in Figure 6A, the stress and elongation ratios at the breaking point of the PTAA-0 (neat PVA) hydrogel were about 86 kPa and 356%, respectively. Both the elongation ratio and stress at breaking point increased when more TA@ANFs were added. A high elongation ratio of 600 ± 8% was achieved for PTAA-25, while its stress at breaking point was only 326 ± 6 kPa. Moreover, because only a small amount of TA@ANFs (0.6~2.8 wt‰) was used as additives in hydrogel preparation, the compressive curves indicated that the PVA/TA@ANFs/Ag hydrogel was very soft, rather than rigid, with the addition of TA@ANFs. As shown in Figure 6B, the compressive stress was almost constant when the hydrogel was subjected to 50% of strain for samples of PATT-5 to PATT-25. The low stress at breaking point as well as the low compressive stress at large deformation demonstrated that the PVA/TA@ANFs/Ag hydrogel was very soft, which is an essential requirement for a strain sensor that would be used on the surface of skin, because large-scale limb movements, or the stretching of skin, may be restricted if a hydrogel is excessively rigid [47]. For a brief understanding of the mechanical performance of PVA/TA@ANFs/Ag hydrogels, we compared the stress and elongation ratio of PTAA-25 with other works in Figure 7. It could be seen that the hydrogel based on tea-stained ANFs, in this work, surpassed most of the other state-of-the-art, PVA-based hydrogels. Therefore, the ultra-stretchable yet soft PVA/TA@ANFs/Ag hydrogel was a very promising material for the preparation of strain sensors.

### 3.4. Performance of the PVA/TA@ANFs/Ag Hydrogels as Wearable Sensors

Ag NP-coated TA@ANFs were critical for the conductivity of PVA/TA@ANFs/Ag hydrogels. We measured the initial electrical resistance (*R*) of the PTAA-0 hydrogel, which was only 0.049 ± 0.004 S/m, and could be considered insulated. Meanwhile, the *R* of PTAA-25 was 1.467 ± 0.007 S/m, because Ag NPs on the surface of TA@ANFs could contact well with each other. This would suggest that a continuous and conductive network was built within the PVA/TA@ANFs/Ag hydrogels. However, this conductive network would be temporarily interrupted when the hydrogel was stretched because insulated gaps would form between two neighboring Ag NPs or TA@ANFs. Therefore, the tensile behavior caused by human bodily motions could be described by the relative variation ratio of the electrical resistance (Δ*R*/*R*_0_) exhibited by PVA/TA@ANFs/Ag hydrogels. As shown in Figure 8A–D, the Δ*R*/*R*_0_ values of the PTAA-5 hydrogel with same dimensions regularly changed with the motion of human limbs when it was pasted onto the surfaces of skin covering various joints. Rapid and stable changes in *R* were observed within 1 s, and Δ*R*/*R*_0_ ratios of 27.5 ± 1.8%, 58.9 ± 8.9%, 17.9 ± 0.6%, and 51.0 ± 1.6% were detected when the bending angles were 90° for the joint of a finger, an elbow, a wrist, and a knee, respectively. It was particularly notable that the hydrogel used on the knee underwent 55.4 ± 1.6% elongation, while its Δ*R* remained stable after many motion cycles. This demonstrated that, in addition to excellent mechanical properties, the PVA/TA@ANFs/Ag hydrogels also had desirable sensitivity for the monitoring of large-scale human limb movements.

### 3.5. Water Absorption and Dewatering Performance of PVA/TA@ANFs/Ag Hydrogels

As a potential strain sensor, the hydrogel’s water absorption and retention performance are critical because sweat will be produced upon human joints as they undergo strenuous motions. Hydrogels that cannot only rapidly absorb moderate water but also readily shed water at room temperature and in humid environments are desirable for applications as strain sensors that could be used outdoors or on the surface of human skin. Therefore, we measured the swelling behavior dehydration kinetics of PVA/TA@ANFs/Ag hydrogels. As illustrated in Figure 9A, in comparison with the PTAA-0 hydrogel, the maximum water absorption was enhanced when TA@ANFs were incorporated into hydrogels, and PTAA-10 showed the highest water absorption ratio, which might be ascribed to the excellent hydrophilicity of the TA layer and the Ag NPs. However, due to the formation of more hydrogen bonds within the interiors of the hydrogels, the maximum water absorption decreased when more TA@ANFs were added. As demonstrated by the swelling kinetics’ measurements (Figure 9B), all of the hydrogels could absorb more than 9.5% of water within a duration of 4 h. The swelling kinetics of PTAA-10 were especially rapid, as it absorbed ~18.9% of water within 30 min. In contrast, the PVA hydrogel could only absorb 3.5% of water during the same time span. These results indicated that the incorporation of hydrophilic TA@ANFs facilitated water absorption by the PVA-based hydrogels. Moreover, as shown in Figure 9C, all of the PVA/TA@ANFs/Ag hydrogels exhibited similar dewatering behavior as that of the PTAA-0 hydrogel, which could lose 90.0% of absorbed water within 2 days.

## 4. Conclusions

In this work, we prepared aqueous dispersions of ANFs and used TA to coat their surfaces via tea-stain chemistry. The aqueous TA@ANFs were further used as an additive to construct PVA/TA@ANFs’ hydrogels. The in situ growth of Ag NPs on the surfaces of the TA@ANFs imparted the hydrogels with electrical conductivity. The resultant PVA/TA@ANFs/Ag hydrogels could be used as sensors to measure the changes of electrical resistance that occur during limb movements and thus to monitor human motion. Due to an abundance of phenols in the TA layer, the interaction between PVA and ANFs was enhanced, and the tensile properties of the hydrogel were significantly improved even though the mass ratio of TA@ANF in the hydrogel was in the range of 0.6~2.8 wt‰. The tensile breaking modulus of the PVA/TA@ANF/Ag hydrogel was in the range of 103–326 kPa, the tensile breaking elongation was in the range of 491–602%, and no obvious deterioration of the flexibility was observed after repeated limb movements. The relative electrical resistance changes (Δ*R*/*R*_0_) of the hydrogel strain sensor were 27.5 ± 1.8%, 58.9 ± 8.9%, 17.9 ± 0.6%, and 51.0 ± 1.6% when the hydrogel was used as a sensor to monitor the joint movements of a finger, an elbow, a wrist, and a knee, respectively. The PVA/TA@ANF/Ag hydrogel could absorb 18.9% of water within 30 min in pure water and release 90.0% of water within 2 days at room temperature and at 60% relative humidity. This ultra-stretchable hydrogel developed in this work was a promising candidate for use in electronic skins or as sensors, especially for the monitoring of large-scale limb motions. The facile strategy described herein may provide a valuable insight for the preparation of hydrogels with high mechanical strength.

## Figures and Tables

**Figure 1 polymers-14-03532-f001:**
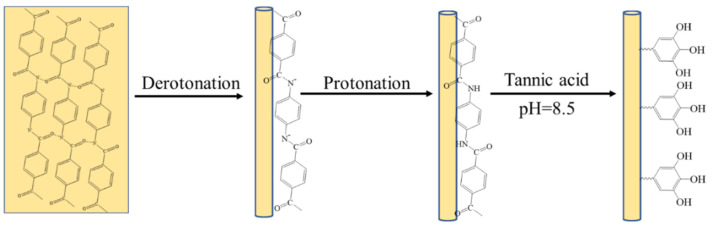
Illustration depicting the preparation of TA@ANFs.

**Figure 2 polymers-14-03532-f002:**
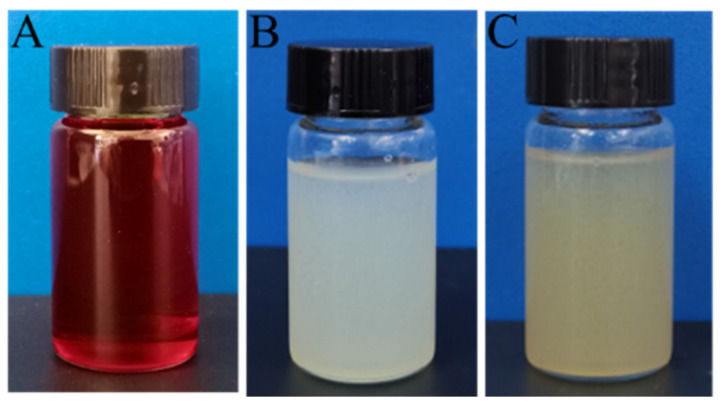
(**A**,**B**) Photographs of ANFs in DMSO (0.2 wt%) and water (0.1 wt%); (**C**) photograph of an aqueous TA@ANFs dispersion (0.1 wt%). All of the dispersions shown in these photographs had been stored for 24 h.

**Figure 3 polymers-14-03532-f003:**
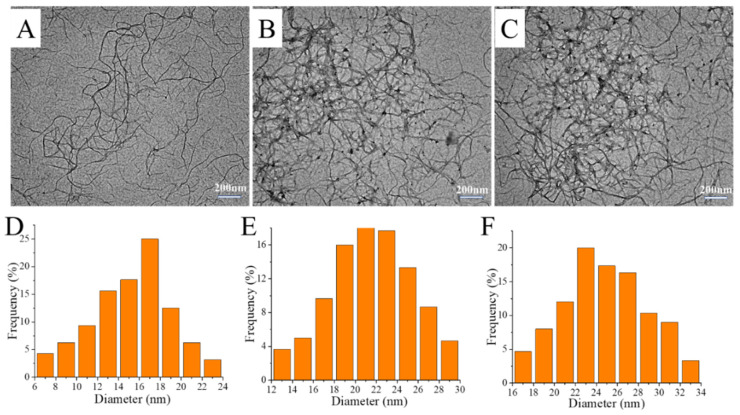
TEM images of deprotonated (**A**) ANFs in DMSO, (**B**) ANFs in water, and (**C**) TA@ANFs in water. Their respective diameter distributions are shown in (**D**–**F**).

**Figure 4 polymers-14-03532-f004:**
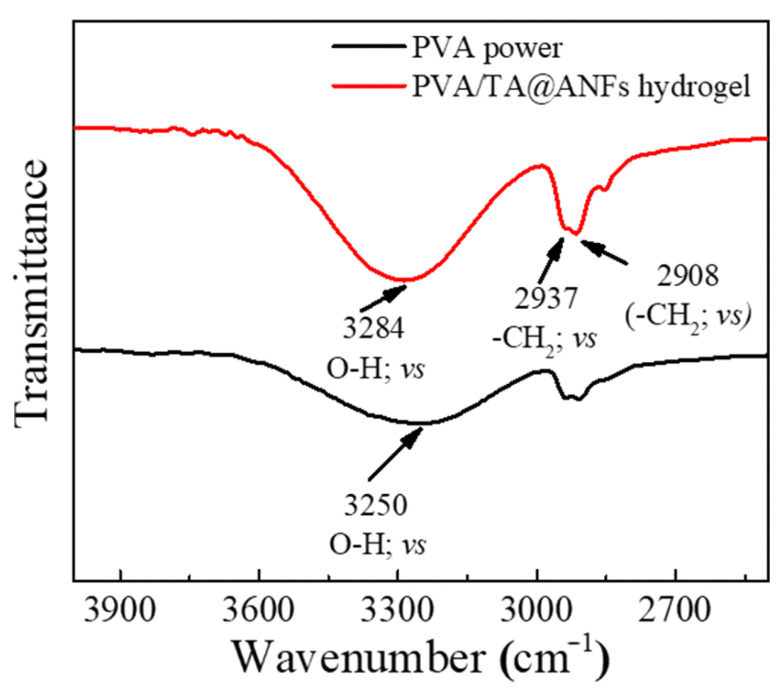
FTIR spectra of PVA powder (black line) and dry PVA/TA@ANFs’ hydrogel (red line).

**Figure 5 polymers-14-03532-f005:**
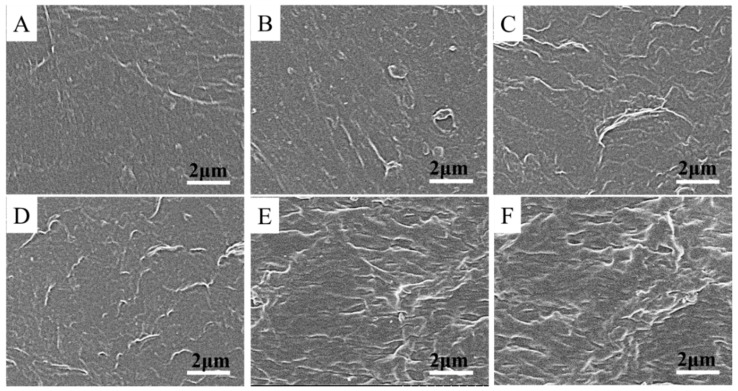
SEM images of PVA/TA@ANFs’ hydrogels with different TA@ANFs’ contents: (**A**) PTAA-0, (**B**) PTAA-5, (**C**) PTAA-10, (**D**) PTAA-15, (**E**) PTAA-20, and (**F**) PTAA-25.

**Figure 6 polymers-14-03532-f006:**
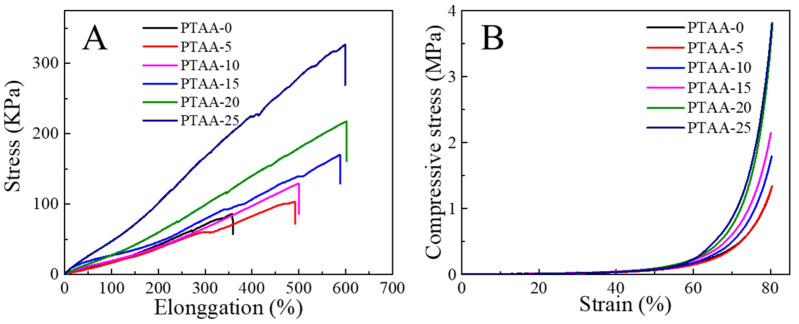
(**A**) The tensile strain–stress curves and (**B**) the compressive strain–stress curves of various PVA/TA@ANFs/Ag hydrogels.

**Figure 7 polymers-14-03532-f007:**
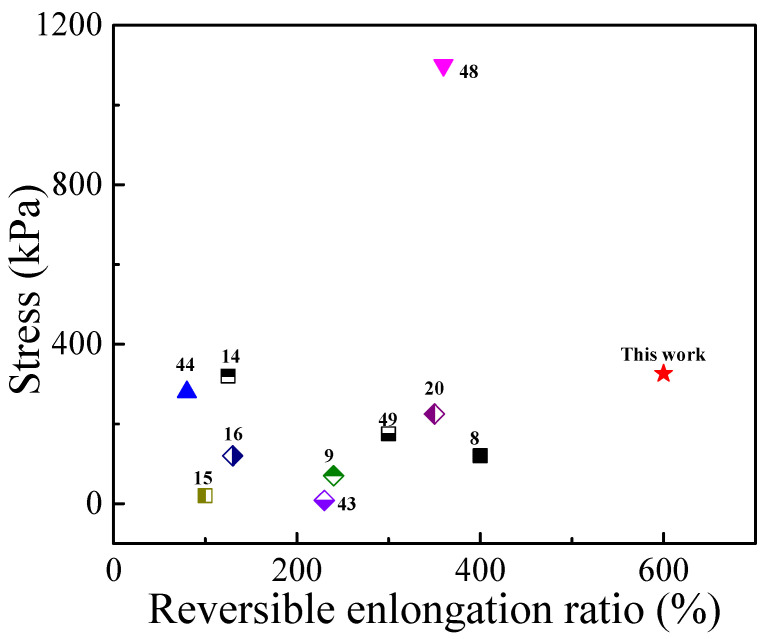
The tensile performance of PTAA-25 in this work and PVA-based hydrogels in others’ reports [8,9,12,14,15,16,21,44,48,49].

**Figure 8 polymers-14-03532-f008:**
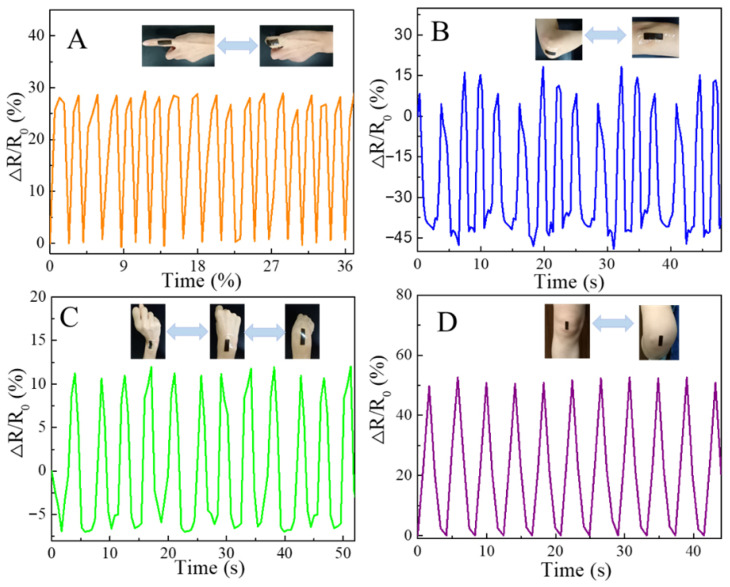
Relative resistance change (Δ*R*/*R*_0_) of PTAA-5 hydrogels on the surfaces of skin covering various joints, including: (**A**) a finger, (**B**) an elbow, (**C**) a wrist, and (**D**) a knee.

**Figure 9 polymers-14-03532-f009:**
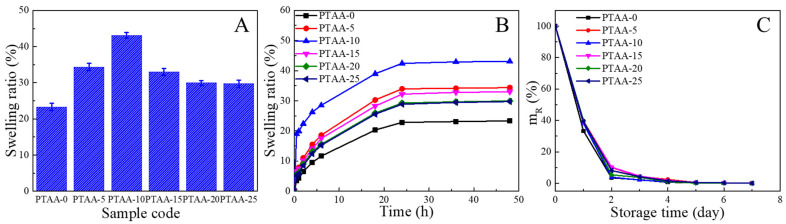
(**A**) Maximum water absorption ratio, (**B**) swelling kinetics, and (**C**) dewatering kinetics of various PVA/TA@ANFs/Ag hydrogels.

**Table 1 polymers-14-03532-t001:** Elemental compositions of neat ANFs and TA@ANFs.

Elemental Compositions	C (%)	N (%)	O (%)
Neat ANFs	72.07	17.77	10.16
TA@ANFs	73.80	13.87	12.33

## Data Availability

Not applicable.

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
