# Peer review of "An Ultra-Stretchable Polyvinyl Alcohol Hydrogel Based on Tannic Acid Modified Aramid Nanofibers for Use as a Strain Sensor"

_polymers, 2022, doi:10.3390/polym14173532_

Round 1
Reviewer 1 Report
Miao et al. performed a chemical and mechanical characterization of modified aqueous PVA solution with TA@ANF/Ag. Good results were reported.
Figure 8. Not explained in the text that sensors used for all joints have the same dimensions.
Line 380: not specified in the text that “neat PA” is PTAA-0 (as reported in Figure 9) and not only aqueous solution.
Well designed experiments, in particular for mechanical tests. However, no biocompatibility tests of the samples were mentioned or performed. Example: in vitro biocompatibility test through the co-culture of hydrogels with cells to evaluate the possible cytotoxicity of the samples.
Author Response
Miao et al. performed a chemical and mechanical characterization of modified aqueous PVA solution with TA@ANF/Ag. Good results were reported.
Figure 8. Not explained in the text that sensors used for all joints have the same dimensions.
In “2.3 Structural Characterization, Sensing measurements”, we have explained this issue, “The ribbon-shaped hydrogel with dimensions of 10 mm ×25 mm × 1 mm was pasted onto the skin covering fingers, elbows, wrists, and knees by adhesive tape.” Maybe this description is inconspicuous, so we emphasized this issue in “3.4 Performance of the PVA/TA@ANFs/Ag hydrogels as wearable sensors” in our revised manuscript. The color of revised sentence was changed into red.
Line 380: not specified in the text that “neat PA” is PTAA-0 (as reported in Figure 9) and not only aqueous solution.
-There is a little of confuse that we did not mention “neat PA” in our manuscript. Maybe reviewers referred here is “neat PVA”? In our manuscript, “neat PVA” referred to “PTAA-0”. To keep the same description in text and Figure 9, we change all “neat PVA” into “PTAA-0”. However, when PTAA-0 appeared first time in manuscript, we still labelled “neat PVA” in brackets, which may be useful for readers to understand.
Well designed experiments, in particular for mechanical tests. However, no biocompatibility tests of the samples were mentioned or performed. Example: in vitro biocompatibility test through the co-culture of hydrogels with cells to evaluate the possible cytotoxicity of the samples.
-Thanks for reviewer’s suggestion. Based on the following reasons, we did not evaluate the ANF/PVA hydrogel’s biocompatibility. The components in ANF/PVA hydrogel are ANFs, PVA, tannic acid, and AgNPs, respectively. PVA is widely used in wound dressing (Carbohydrate Polymers, 2012, 90, 658). Aramid fiber is highly stable, which can be used in bullet-proof vest and worn onto soldier’s body; m-aramid even showed antimicrobial activity (Industrial & Engineering Chemistry Research, 2011, 50, 8693). The leakage of AgNPs from hydrogel may have risk for human skin, but the ANF/PVA hydrogel was not touch with skin, adhesive tape was used to help the hydrogel fixed onto the surface of skin. Meanwhile, due to the strong complexation effect of tannic acid and PVA on Ag atom, the leakage risk of AgNPs is very low. Tannic acid existed in lots of foods, such as tea, red wine, and persimmon.
Reviewer 2 Report
this is a study on the synthesis of PVA-based hydrogels by freeze-thaw and their use as strain sensors.
-The main components of tea are polyphenolic structures such as caffeine and catechin, but the main ingredient is not tannic acid. With the use of tannic acid to modify ANF, the word "...tea-strain" in the title(..Polyvinyl Alcohol Hydrogel Based on Tea Stain-Inspired Aramid Nanofibers for use as a Strain Sensor) was not appropriate.
fig 1 and fi2 have to be separated with text. fig3 and table 1 have to be separated with text. fig6 and fig 7 have to be separated with text.
A similar study involved PVA was in literature. compare the results. doi.org/10.1038/s41598-020-77139-2
In another study, Various uses of hydrogels which included PVA and TA, have been mentioned.. Add the reference. doi.org/10.3390/polym14010070
In this article, only freeze-thaw 3 was used in the synthesis. For example, did you have any observations about freeze-thaw 1, or 2? If you didn't use it, why did you choose 3?
Author Response
This is a study on the synthesis of PVA-based hydrogels by freeze-thaw and their use as strain sensors.
-The main components of tea are polyphenolic structures such as caffeine and catechin, but the main ingredient is not tannic acid. With the use of tannic acid to modify ANF, the word "...tea-strain" in the title (..Polyvinyl Alcohol Hydrogel Based on Tea Stain-Inspired Aramid Nanofibers for use as a Strain Sensor) was not appropriate.
-We have changed the title of the manuscript into “An Ultra-Stretchable Polyvinyl Alcohol Hydrogel Based on Tannic Acid modified Aramid Nanofibers for use as a Strain Sensor”.
Fig 1 and Fig 2 have to be separated with text. Fig3 and Table 1 have to be separated with text. Fig 6 and Fig 7 have to be separated with text.
-All the neighboring figures and tables in manuscript were separated as the reviewer’s requirements
A similar study involved PVA was in literature. Compare the results. doi.org/10.1038/s41598-020-77139-2
-There are four major differences between our manuscript and the article from Mubarak et al.
First, we have reported a hydrogel based strain sensor in our work, while Mubarak et al. have reported a buckypaper based strain sensor. Hydrogel belongs to 3D materials, while buckypaper belongs to 2D material.
Second, the conductive filler was graphene in Mubarak’s work, while the conductive filler in our work was Ag nanoparticle (AgNPs) that in situ formed on the surface of tannic acid (TA) modified aramid nanofibers (ANFs).
Third, the interfacial compatibility between PVA and nanofillers, whatever for graphene or aramid nanofiber, should be enhanced. The surface oxidation facilitates the formation of polar groups on graphene, but ANF, as polymeric material, cannot be treated via the same method. In our work, TA is used because its deposition conditions is mild, large amount of polar -OH group could be induced on the surface of ANF.
Forth, we have to admit that the sensitivity of buckypaper based strain sensor is much higher than ANF/PVA hydrogel based strain sensor. Graphene is a better conductive filler than AgNPs. However, the composited ANF/PVA hydrogel in our work have excellent mechanical properties.
In another study, various uses of hydrogels which included PVA and TA, have been mentioned. Add the reference. doi.org/10.3390/polym14010070
-This reference have been cited in appropriate locations. The citation in our revised manuscript was colored by red.
In this article, only freeze-thaw 3 was used in the synthesis. For example, did you have any observations about freeze-thaw 1, or 2? If you didn't use it, why did you choose 3?
-Base on naked eyes observation, the ANFs/PVA hydrogel treated with only 1 time of “freeze-thaw” cycle was fluidic, cannot be used as strain sensor. The hydrogel treated with 2 times of “freeze-thaw” cycle showed poor mechanical properties, which could be easily broken to pieces just by finger. The hydrogel treated with 3 times of “freeze-thaw” cycle has desirable mechanical properties. We also found that the mechanical properties of hydrogel treated with 3 times of “freeze-thaw” cycle, would not be further improved when more times of “freeze-thaw” cycle was used. These phenomenon were added in our revised manuscript.